# Predicting Keratoconus Progression and Need for Corneal Crosslinking Using Deep Learning

**DOI:** 10.3390/jcm10040844

**Published:** 2021-02-18

**Authors:** Naoko Kato, Hiroki Masumoto, Mao Tanabe, Chikako Sakai, Kazuno Negishi, Hidemasa Torii, Hitoshi Tabuchi, Kazuo Tsubota

**Affiliations:** 1Department of Ophthalmology, School of Medicine, Keio University, Tokyo 160-8582, Japan; QVF03264@nifty.com (C.S.); kazunonegishi@keio.jp (K.N.); hidemasatorii@yahoo.co.jp (H.T.); tsubota@z3.keio.jp (K.T.); 2Department of Technology and Design Thinking for Medicine, Graduate School of Biomedical Sciences, Hiroshima University, Hiroshima 734-8551, Japan; hiroki.masumoto.business@gmail.com (H.M.); m.tanabe@tsukazaki-eye.net (M.T.); H.Tabuchi@tsukazaki-eye.net (H.T.); 3Tsubota Laboratory, Inc., Tokyo 160-0016, Japan

**Keywords:** keratoconus, progression, deep learning, prediction, corneal crosslinking, tomography, patients’ age

## Abstract

We aimed to predict keratoconus progression and the need for corneal crosslinking (CXL) using deep learning (DL). Two hundred and seventy-four corneal tomography images taken by Pentacam HR^®^ (Oculus, Wetzlar, Germany) of 158 keratoconus patients were examined. All patients were examined two times or more, and divided into two groups; the progression group and the non-progression group. An axial map of the frontal corneal plane, a pachymetry map, and a combination of these two maps at the initial examination were assessed according to the patients’ age. Training with a convolutional neural network on these learning data objects was conducted. Ninety eyes showed progression and 184 eyes showed no progression. The axial map, the pachymetry map, and their combination combined with patients’ age showed mean AUC values of 0.783, 0.784, and 0.814 (95% confidence interval (0.721–0.845) (0.722–0.846), and (0.755–0.872), respectively), with sensitivities of 87.8%, 77.8%, and 77.8% ((79.2–93.7), (67.8–85.9), and (67.8–85.9)) and specificities of 59.8%, 65.8%, and 69.6% ((52.3–66.9), (58.4–72.6), and (62.4–76.1)), respectively. Using the proposed DL neural network model, keratoconus progression can be predicted on corneal tomography maps combined with patients’ age.

## 1. Introduction

The first human study of the corneal crosslinking (CXL) to halt the progression of keratoconus/keratectasia was reported by Wollensak et al. [1] in 2003, at the time thought to be an incurable disease. Patients with this condition sometimes had to endure pain when wearing contact lenses, with the sudden occurrence of acute hydrops as an additional complication. Keratoplasty has been necessary with disease progression in some cases.

The primary purpose of CXL is to halt the progression of keratoconus. The best candidate for CXL is an individual with keratoconus or post-refractive surgery ectasia who has recently revealed disease progression. However, there are no definitive criteria for predicting keratoconus progression at present. The parameters that must be considered are changes in refraction (including astigmatism), uncorrected and best spectacle-corrected visual acuities, and corneal shape and thickness (according to corneal topography or tomography) [2,3,4,5,6].

Widely accepted indications for CXL include an increase of 1.00 diopter (D) or more in the steepest keratometry measurement, an increase of 1.00 D or more in the manifest cylinder, and an increase of 0.50 D or more in the manifest refraction spherical equivalent in 12 months [7]. It may take several months to years to determine whether a patient meets the clinical criteria for CXL. However, especially for some patients, the disease may exacerbate rapidly during the follow-up period, even while awaiting CXL [8]. Therefore, a method for predicting the progression and the need for CXL in keratoconus cases at the first examination is required.

Artificial intelligence (AI) is the fourth industrial revolution in mankind’s history, and deep learning (DL) is a class of state-of-the-art machine learning techniques that has sparked tremendous global interest in recent years [9]. In the field of ophthalmology, DL use for the diagnosis of diabetic retinopathy, glaucoma, age-related macular degeneration, and retinopathy of prematurity using fundus photographs and/or optical coherence tomography (OCT) have been developed [10,11,12,13,14,15,16]. For corneal diseases, DL can predict the likelihood of the need for future keratoplasty treatment [17]. Recently, DL has been used for the detection and staging of keratoconus [18,19,20]; however, the ability of DL to predict progression, namely the decision for CXL indication, has not been reported. 

In the present work, we aimed to determine the need for CXL to halt keratoconus progression using DL. To our knowledge, this is the first trial to distinguish the indication for CXL by DL.

## 2. Materials and Methods

This study followed the ethical standards of the Declaration of Helsinki and the study protocol was approved by the Institutional Review Board of the Keio University School of Medicine.

We retrospectively analyzed the axial and the pachymetry maps combined with the patients’ age at the initial visit of each patient by DL. Two hundred and seventy-four eyes of 158 patients with keratoconus (112 males and 46 females; mean age, 27.8 ± 11.7 years), who visited the Department of Ophthalmology, Keio University School of Medicine from January 2009 to August 2018 at least twice, were included to the present study and retrospectively examined (Appendix A). Tomography images of those eyes were taken using Pentacam HR^®^ instrument (Oculus, Wetzlar, Germany) by trained certified ophthalmic technicians at the first visit (Appendix A). Keratoconus was diagnosed based on corneal tomography, i.e., ectasia screening using the CASIA^®^ device (Tomey, Aichi, Japan), and/or topographic keratoconus classification using the Pentacam HR instrument. Eyes with pellucid marginal degeneration, keratectasia after laser refractive corneal surgery, previous acute hydrops, or other ocular surface diseases were excluded. Then, the patients were followed 2 times or more with certain intervals. The mean period between the initial and final examination was 2.60 ± 2.09 years (varied from 6 weeks to 8.6 years).

CXL treatment was applied to eyes with recently active keratoconus that showed significant keratoconus progression, based on aforementioned criteria by corneal specialists. Eyes that underwent CXL were assigned to the progression group; eyes that did not undergo CXL were placed in the non-progression group (Figure 1).

We created an AI model to predict conical cornea progression with an axial map (Axial), a pachymetry map (Pachy), and their lateral combination (Both) taken at the first visit, using a Pentacam HR instrument; the assessments were based on patients’ age (Figure 2).

The K-fold (K = 5) cross-validation method [21,22] was used in this study. The original sample was randomly partitioned into k subsamples. K-1 subsamples were used as training data after data augmentation and the remaining single subsample was retained as the validation data for testing the model. The cross-validation process was repeated k times, with each of the subsamples was used as the validation data. All images were resized to 224 pixels × 224 pixels.

The deep neural network model was constructed based on the Visual Geometry Group-16 (VGG-16) [23,24,25]. The five blocks with convolutional layers, rectified linear unit activation function and max pooling layer of the VGG-16 [26,27,28] were used in this neural network. We used parameters from ImageNet blocks 1–4. This method is called fine-tuning and used in various studies [29].

After five convolutional blocks, the global average pooling layer is passed, such that spatial information is removed from the extracted features. After the global average pooling layer, we combined the standardized age information. The ratio of the amount of age information to the amount of image information is referred to here as the parameter ratio, described below. The extracted features were then compressed by passing through the fully connected layers. The last fully connected layer with the activation function, Softmax, evaluated the probability of each class (i.e., the two groups comprising the progression group and the non-progression group). The number of units in the hidden layer (n_dim) is described below.

We used the optimization momentum stochastic gradient descent algorithm (inertial term = 0.9) [30,31] as the optimizer. The learning ratio of the optimizer is described below (Figure 3).

The images were compressed in five blocks of visual geometry group-16 network and a global average pooling layer. Afterwards, standardized age information was combined in the “parameter ratio”. The extracted features were then compressed by passing through the fully connected layers. The last fully connected layer with the activation function, Softmax, evaluated the probability of each class (i.e., the two groups comprising the progression group and the non-progression group). VGG-16: Visual Geometry Group-16 and n_dim, the number of units in the hidden layer.

The parameter ratio, n_dim, and learning ratio were chosen from a uniform distribution from 0.2 to 0.8, an exponential distribution from 26 to 28, and a logarithmic distribution from 10-4 to 10-2, respectively. The performance of our approach was evaluated using the k-fold cross-validation method 10 times. The parameters with the highest area under the curve (AUC) were used. The developed prediction model and training were applied using Python TensorFlow (https://www.tensorflow.org/ (accessed on 15 February 2021)). We used Optuna (https://optuna.readthedocs.io/en/stable/index.html (accessed on 15 February 2021)) for setting the hyperparameter. The training and analysis codes are provided in Dataset S1.

The performance metrics were AUC, sensitivity, and specificity. The receiver operating characteristic curve (ROC) and the AUC were calculated using the NN’s output as the probability that a certain image belonged to the progression group, in addition to actual progression information. Using the Youden index [32] in the ROC curve, we defined the optimal cutoff value and the sensitivity and specificity of the cutoff value.

We compared patient age with the Welch’s *t*-test and male–female ratio with the Fisher’s exact test. *p* < 0.05 was considered statistically significant. Statistical analysis was performed using Python Scipy (https://www.scipy.org/ (accessed on 15 February 2021)) and Python Statsmodels (http://www.statsmodels.org/stable/index.html (accessed on 15 February 2021)).

For the AUC analysis, the AUC was assumed to be normally distributed and the 95% confidence interval (CI) was calculated with the following formula:95%CI=AUC±Z(0.975)∗SE(AUC)
$$Z\left(x\right)=\frac{1}{\sqrt{2\pi }}exp\left(-\frac{x^{2}}{2}\right)$$
$$SE\left(AUC\right)=\sqrt{\frac{AUC\left(1-AUC\right)+\left(n_{p}-1\right)\left(Q_{1}-AUC^{2}\right)+\left(n_{N}-1\right)\left(Q_{2}-AUC^{2}\right)}{n_{p}n_{N}}}$$
$$Q_{1}=\frac{AUC}{2-AUC}$$
$$Q_{2}=\frac{2AUC^{2}}{1+AUC}$$

## 3. Results

### 3.1. Background

Ninety eyes showed progression and were included in the progression group; the other 184 eyes did not show progression and were placed in the non-progression group. The background information of both groups is listed in Table 1.

### 3.2. Evaluation of Keratoconus Progression

The predictive performance of keratoconus progression is shown in Table 2 and Figure 4.

## 4. Discussion

In this study, we developed a new method to predict the progression of keratoconus using DL via an AI platform. When the possibility for keratoconus progression was combined with patients’ age, the AUC values were 0.783 (0.721–0.845) with the axial map, 0.784 (0.722–0.846) with the corneal pachymetry map, and 0.814 (0.755–0.872) using both maps.

The age of the enrolled patients was significantly younger in the progression group than in the non-progression group. Age is an inevitable factor in keratoconus progression, partly because keratoconus is a disorder that tends to progress depend on the patients’ age; its progression tends to slow during middle age but young onset keratoconus has been shown to progress much faster [33,34]. We had previously investigated the condition of keratoconus patients who were followed-up twice or more after the initial visit and found that the patients’ age was the most relevant factor with respect to keratoconus progression, followed by Rmin (the minimum sagittal curvature evaluated by Pentacam HR) of the corneal frontal plane [35]. The disproportionate influence of age between progression and non-progression groups was ineluctable.

Considering that a young age is relevant to keratoconus progression, we applied patients’ age to corneal tomography data to predict progression using our DL approach. We used three types of maps: an axial map of the corneal frontal plane, a pachymetry map, and a combination of the two; the three map types showed similar AUC, sensitivity, and specificity values. This demonstrates the clinical versatility of DL for predicting the progression of keratoconus, as the axial map of the corneal frontal plane is usually displayed by every corneal topography/tomography device.

An AUC value of 0.78–0.81 is not a perfect indicator for progression prediction. For keratoconus specialists, who determine empirically the indication of CXL considering the clinical stage of keratoconus and patients’ age, the diagnosis rate may not be sufficient; however, it could serve as an indicator for non-specialists such as family practitioners, general ophthalmologists/optometrists, or specialists in other fields of ophthalmology to help them decide whether patients should consult with corneal specialists trained in CXL.

In the present investigation, we used the images of the corneal tomography resized to 224 pixels × 224 pixels. This compression could be suitable for the analysis or corneal tomography/topography, because the present investigation requires analysis of relatively large area of the pictures, not like finding out the tiny hemorrhages like in the assessment of diabetic retinopathy. This character can help accelerate the calculation speed and also be applied to other devices with lower specification, possibly bringing a great advantage for the clinical use.

This is the first trial that proposes the prediction of keratoconus progression by corneal topographic data at the first visit of the patients using DL as long as our knowledge, containing some limitations. The limitations of the present study included relatively small number of the participants and variation in the follow-up period to determine the indication for CXL (from 6 weeks to 8.6 years). Reassessment of all cases followed up for more than 2 years may provide a more accurate representation.

We excluded cases with pellucid marginal degeneration from the present study, which shows protrusion and thinning of the lower part of the cornea that occurs after the third or fourth decade of life and continues to progress even after middle-age. This was done partly because we thought that this condition might be different from the usual keratoconus and also because the number of patients with pellucid marginal degeneration was too small (less than 10% of the whole). The mechanisms for delayed occurrence and progression of protrusion in eyes with pellucid marginal degeneration has not been clarified. DL using a large number of keratoconus cases may elucidate these unclarified questions.

## 5. Conclusions

We attempted to predict the exacerbation of keratoconus that required CXL aftertime using DL and showed that the axial map or the pachymetry map combined with the patients’ age were useful indicators of the need for CXL, with about 80% probability. The prediction of keratoconus progression using AI-based DL is expected to help ophthalmologists/optometrists, and especially non-specialists, regarding the timing of referring patients to corneal specialists for CXL treatment. 

## Figures and Tables

**Figure 1 jcm-10-00844-f001:**
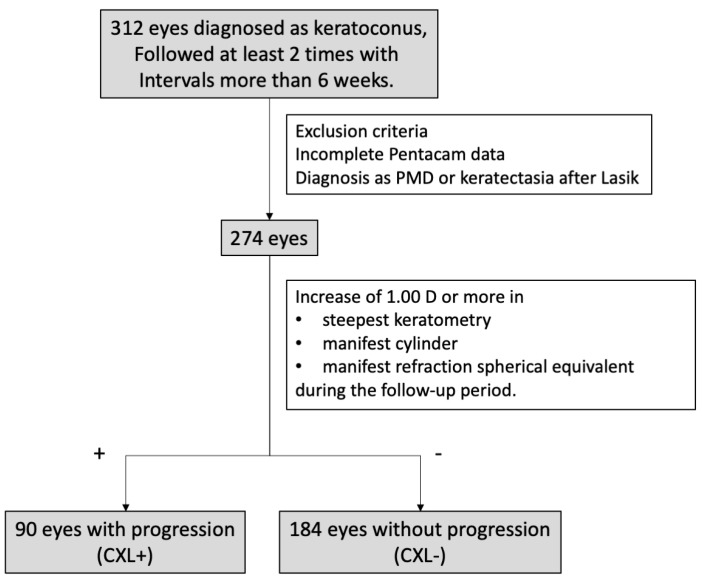
The flowchart representing the selection of enrolled patients. CXL: corneal crosslinking.

**Figure 2 jcm-10-00844-f002:**
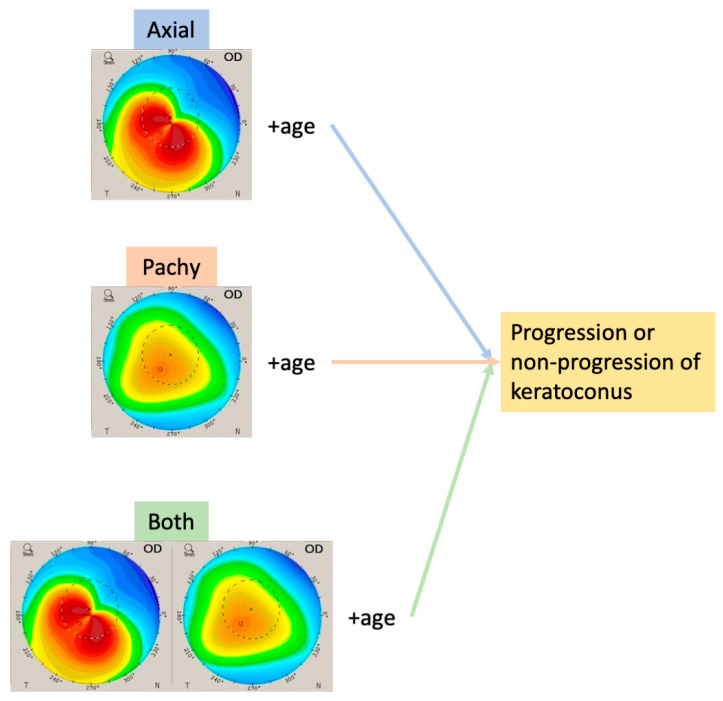
Conical cornea progression prediction using a combined axial map, pachymetry, and a combined image of the two, with respect to patients’ age.

**Figure 3 jcm-10-00844-f003:**
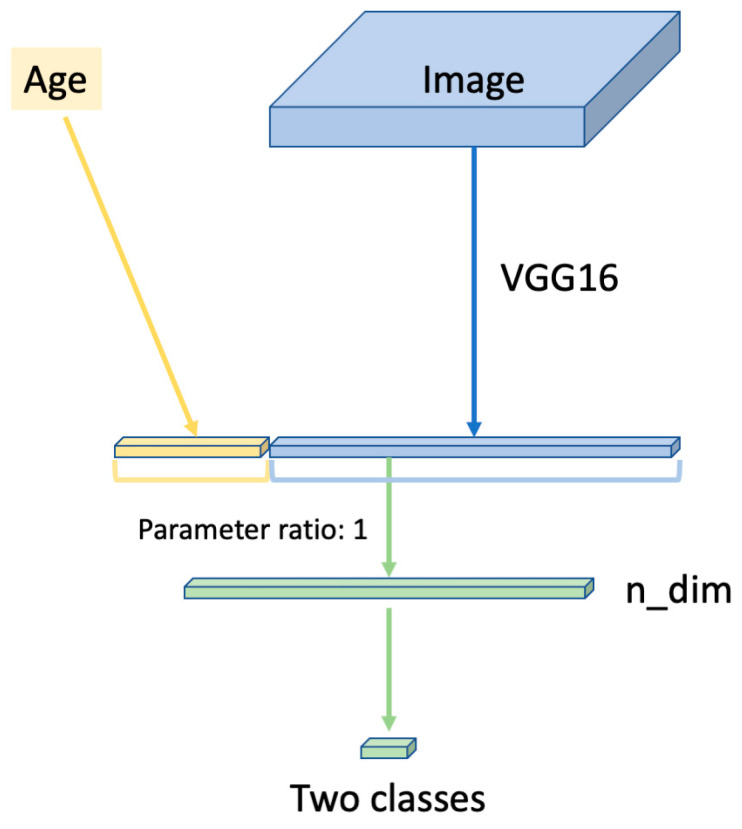
Overall architecture of the neural network model used in this study. VGG: Visual Geometry Group.

**Figure 4 jcm-10-00844-f004:**
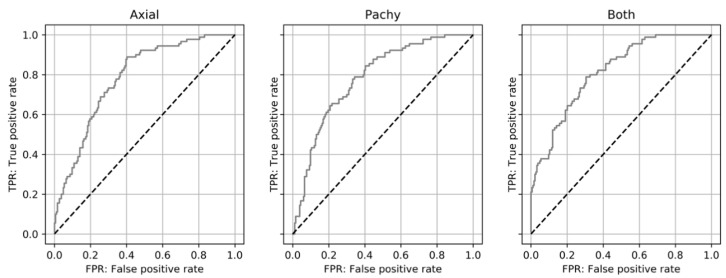
Receiver operating characteristic (ROC) curve of the probability for cross-linking (CXL) among keratoconus eyes. Axial map (Axial), pachymetry map (Pachy), and a combination of the two maps (Both) reveal a similar ROC curve of probability for CXL.

**Table 1 jcm-10-00844-t001:** Patients’ background.

	Progression Group	Non-Progression Group	*p*-Value
Age (mean ± SD)	21.0 ± 5.9	31.5 ± 12.4	*p* < 0.01
Gender (female ratio)	24/90	55/184	*p* = 0.67

SD: standard deviation.

**Table 2 jcm-10-00844-t002:** Area under the curve (AUC), sensitivity, and specificity outcomes obtained by assessment using an axial map, a corneal pachymetry map, and a combination of the two, with or without respect to patients’ age.

	AUC	Sensitivity	Specificity
Axial	0.783 (0.721–0.845)	87.8% (79/90) (79.2–93.7)	59.8% (110/184) (52.3–66.9)
Pachy	0.784 (0.722–0.846)	77.8% (70/90) (67.8–85.9)	65.8% (121/184) (58.4–72.6)
Both	0.814 (0.755–0.872)	77.8% (70/90) (67.8–85.9)	69.6% (128/184) (62.4–76.1)

AUC, area under the curve.

## Data Availability

The data presented in this study are available in Appendix A.

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
