# Peer review of "Predicting Keratoconus Progression and Need for Corneal Crosslinking Using Deep Learning"

_jcm, 2021, doi:10.3390/jcm10040844_

Round 1
Reviewer 1 Report
Thank you for consulting me to review this interesting manuscript. In this paper the authors develop a DL model to evaluate the need for CXL in patients with keratoconus. I have the following concerns with this manuscript:
- The authors state in the introduction that keratoplasty was ‘often’ required. This is inaccurate. Keratoplasty has typically been required in a minority of keratoconus patients. Most patients are managed well with spectacles or contact lenses. This should be amended.
- Within the introduction, the authors state the parameters used to define progression but do not provide a timeline. Are the changes in keratometry, etc. described over 1-year? 6-months? The established criteria for progression within the literature is a change in the Kmax on topography of 1.0D or more within 12-months. It is advisable the authors further narrow their criteria for progression so as to avoid inter-visit fluctuations within corneal bioparameters.
- The authors incorrectly state in the discussion that keratoconus progresses ‘because of poverty’. This should be removed.
- The authors should provide further discussion of factors which are linked to keratoconus progression. For instance, younger patients typically have more aggressive disease, this should be clarified. The reasons behind this remain unclear and are likely multifactorial, stemming from increased incidence of atopy, and increased preponderance to eye-rubbing.
- The authors do not clearly describe how their DL model was used. Was it used to retrospectively evaluate patients who had already gone through their pipeline of treat versus do not treat? Was it implemented prospectively in their decision-making? This needs to be clarified further within the methods and the introduction so it is apparent how the DL model was developed and implemented.
- Although an AUC value of 0.78 is excellent, it is far from perfect and the authors should be cautious in discussing implementation. This is an excellent pilot study but it should be recognized as such.
- The manuscript requires thorough copy-editing and proofreading as there are many grammatical mistakes throughout.
Reviewer 2 Report
The authors do a good introduction on the possible use of AI in Ophthalmology in predicting keratoconus progression and need for corneal CXL.
The paper is generally well-written; however, the material and method section may be not easily clear for readers who are unfamiliar with AI models, deep learning and neural networks (cross-validation process…). I believe that a brief, simple basic explanation of the methods can be useful.
The aim of this study seems to be to assess the accuracy of the deep learning model created by authors in the prediction of keratoconus progression.
The sensitivity and the specificity of the model, as remarked by authors in the conclusion section, are not satisfactory for keratoconus specialists, but may be sufficient for non -specialists: on the contrary, in my opinion, due to its low sensitivity this model can be potentially dangerous for non-specialists as well, giving around 20% of false negatives, who are potentially lost to the follow-up and miss an appropriate treatment (CXL). Therefore, the model should be further improved in order to have a better accuracy, especially in terms of sensitivity.
This important limitation of the model should be better underlined in the discussion.
The model created by the authors consider only anterior axial map, pachymetry map (with their lateral combination) and age: the authors should explain why only these two maps were taken into account (why not posterior maps?
Round 2
Reviewer 2 Report
The specifications enclosed in the answers to reviewers clarify the significance and the findings of the work.
This manuscript is a resubmission of an earlier submission. The following is a list of the peer review reports and author responses from that submission.